# Contradictions and possibilities for change: Exploring stakeholder perspectives of Canada's Feminist International Assistance Policy (FIAP) and their connection to a future for global health

**Hanna Chidwick**[1]*, **Germaine Tuyisenge**[2], **Deborah D. DiLiberto**[1], **Lisa Schwartz**[3]

**1** Faculty of Health Sciences, Global Health, McMaster University, Hamilton, Ontario, Canada, **2** Faculty of Health Sciences, Simon Fraser University, British Columbia, Canada, **3** Faculty of Health Sciences, Health Research Methods, Evidence and Impact, McMaster University, Hamilton, Ontario, Canada

* chidwihw@mcmaster.ca

**Data Availability Statement:** The relevant data underlying this study are included within the article and its supporting information files. De-identified

## Abstract

Increasing global inequities have led to ongoing critiques of both the historical roots and current practices in global health. From this literature, questions have emerged about the future of global health and Canada's role in this future. However, there is little research exploring the role of Canadian policy for global health funding and the experience of stakeholders that currently implement projects with Canadian funding. This study explores stakeholder perspectives of how Canada's Feminist International Assistance Policy (FIAP) influences adolescent sexual and reproductive health (ASRH) projects and how these perspectives align with calls for change in global health as proposed by Chidwick et al. in the conceptual framework for an imagined future for global health. This study was conducted from February to May 2023 through eight interviews with key informants who were working on ASRH projects, funded through Global Affairs Canada. The study approach included stakeholder analysis and qualitative description. Results highlight the role of Canadian organizations in ASRH projects, importance of partnerships, influence of the FIAP, operational and contextual tensions that arise from implementing FIAP values in ASRH projects outside of Canada, along with stakeholder views on moving forward in global health policy and practice. Findings indicate that the language of the FIAP is aligned with calls for change in global health although there is opportunity for further action towards operationalizing this change. Specifically, findings highlight opportunity to create more flexible funding processes, expand monitoring and evaluation approaches to better include feminist approaches, promote rights-based and dignity-based practices in ASRH, and increase timelines to facilitate greater consultation and partnership building with communities. In conclusion, the FIAP creates an impactful foundation for change towards equity and justice in ASRH projects, although Canadian funding structures and processes need to continue to re-imagine how they support organizations to action these changes.

raw data that further support the findings are available from the Hamilton Integrated Research Ethics Board (HiREB) upon reasonable request. To request access, please contact Mark Inman (inmanma@mcmaster.ca), HiREB, 293 Wellington St. N., Suite 120, Hamilton, ON L8L 8E7, Canada. Telephone: 905-521-2100, Ext. 42013; Fax: 905-577-8378.

**Funding:** The authors received no specific funding for this work.

**Competing interests:** The authors have declared that no competing interests exist.

# Introduction

More than half of the global population is comprised of adolescents, defined by the United Nations (UN) as young people aged 10 to 19 years [1, 2]. This period of development is pivotal in shaping both individual and community health outcomes [1, 2]. Adolescents may start engaging in sexual behaviour around the age of 15 years making their sexual and reproductive health (SRH) an important issue for both their immediate and long-term health outcomes [1, 3]. SRH is defined as,

> A wide range of health issues including family planning; maternal and newborn health care; prevention, diagnosis and treatment of sexually transmitted infections (STIs) . . . [with] services [aiming to prevent] poor SRH, such as . . . unintended pregnancies, unsafe abortions, [and] complications caused by STIs. [4, 5]

Improving SRH outcomes aligns with Sustainable Development Goals (SDG) 3.7 and 5.6, which aim to ensure universal access to SRH services and rights [4, 5]. In sub-Saharan Africa, and Eastern sub-Saharan Africa in particular, advancements towards SDG targets remain slow, resulting in persistent challenges with STIs, unmet contraceptive needs, and inadequate quality of SRH care among adolescents [4, 6–13]. For example, by the age of 19, over half of adolescent girls in Tanzania have given birth to a child indicating limited access to SRH information and services [14]. As such, adolescent SRH (ASRH) has become the focus of many foreign governments through development assistance.

Over the last six years, Canadian international development assistance has increasingly prioritized ASRH in sub-Saharan Africa in comparison to other global health initiatives [15]. In 2017, the Canadian government announced the Feminist International Assistance Policy (FIAP), which guides Canadian funding for development abroad [16]. The FIAP encourages a specific focus on gender equality and human dignity, including SRH [16]. After implementing the FIAP, the government announced $650 million over three years to address gaps in SRH globally, where 95% of all financial support for development initiatives would target and integrate gender equality and empowerment of women and girls by 2022 [16–19]. In 2019 and a further part of Canada's FIAP, the Liberal government announced a "10-Year Commitment to Global Health and Rights" investing an additional $1.4 billion each year (for 10 years) to further support the health and rights of women and girls globally [17]. This included a commitment of $700 million each year specifically for SRH [17]. Almost 50% of projects funded through the FIAP focus in sub-Saharan Africa, where many non-governmental organizations (NGOs) working in Eastern sub-Saharan Africa in particular, focus on ASRH [16, 20–22].

Global Affairs Canada (GAC) under the leadership of the current government, leads the implementation of the FIAP by distributing funds to relevant projects [15, 23]. Significant to this distribution of funds are organizations, as the FIAP promotes multi-sectoral approaches by engaging global and local civil society, multinational organizations, women's rights organizations, and non-traditional donors [16]. Over the past decades, these various NGOs have grown into one of the most significant groups of actors influencing development and health globally [24–26]. Since 2018, GAC has focused its funding on projects that adopt a feminist and sustainable approach to improve the health, rights, and well-being of adolescent girls, women, and children [27].

## Study purpose & rationale

Several studies have examined the FIAP including stakeholder perceptions of feminism within the policy and challenges with its implementation [19, 20, 28–33]. These studies, along with

GAC, have established NGOs as important and prominent actors in implementing FIAP funded projects [34], specifically for ASRH. However, there has been limited exploration of how the FIAP practically impacts ASRH projects that are implemented by NGOs, and further, what this impact indicates about the future of ASRH and global health generally. This study aims to understand how the FIAP influences ASRH projects and examine how this aligns with the conceptual framework for an imagined future for global health, as proposed by Chidwick et al. [35].

This study examines the influence of the FIAP through GAC processes at different stages of the project (i.e., development, implementation, and evaluation) and in different aspects of the project (i.e., project values, approaches to ASRH and partnerships, integration of gender, project accountability and reporting structures). Exploring how the FIAP influences projects and work by NGOs is not a new discussion as evidenced by scholarship in the development sector which has examined the FIAP's influence in how organizations are addressing gender equality targets [36].

## Research questions & study objectives

The research question for this study asks, *how does the FIAP influence ASRH projects in Eastern sub-Saharan Africa and how do these perspectives relate to an imagined future for global health*? The study objectives include the following:

1. To describe key informant perspectives about how the FIAP influences ASRH projects in Eastern sub-Saharan Africa.

2. To examine how the FIAP and its connection to ASRH projects aligns with the conceptual framework for an imagined future for global health, as proposed by Chidwick et al. [35].

## Conceptual framework: An imagined future for global health

In this study, the FIAP is analyzed in relation to a previously developed conceptual framework for an imagined future for global health as proposed by Chidwick et al. [35]. In the past few years, there has been growing literature and discussion concerning the historical roots and current practices in global health [37]. Many of these critiques concern the future of the field and specifically, how global health and its actors need to change to redress past transgressions and move towards greater equity and justice in policy, research, and practice [38–51]. Based on scholarship from Futures Studies [52, 53], which offers a process to conceptualize multiple different futures, the conceptual framework was developed through an analysis of the history and contemporary critiques of the field. This analysis prioritized the inclusion of literature written by scholars who have historically and continue to be racialized and/or marginalized in their opportunity to lead in the field. The analysis also prioritized critical reflexivity to identify the biases, assumptions, privilege and power amongst the authorship team that influenced the analysis of literature.

The framework presents one potential alternative to the present context and processes in global health research, policy and practice. The conceptualization of this imagined future does not aim to evaluate or predict the future of global health as a field, but rather explore one possible vision of change as it relates to ASRH policy, research, and practice [37, 39, 44, 47, 48, 52, 54–57]. The framework describes two overarching shifts in this potential future, first, shifting the power in how we do global health and second, shifting the paradigm in which we think about global health. These shifts are practically conceptualized in terms of funding, leadership, knowledge production, knowledge history, knowledge justice, and reflexivity (see Table 1).

**Table 1. Conceptual framework.**

| An imagined future for global health | | |
|---|---|---|
| Shift power in how we **do** global health | **Funding** | • Redistribution and decentralization of resources to local (i.e., people closest to the work, communities of focus while acknowledging context specificity of the "local")<br>• Funder accountability to local<br>• Diversification of decision making to include local (collaboration)<br>• Better align funding with local priorities |
| | **Leadership** | • Institutions reorient leadership to local, decentralize decision making power<br>• South-South collaboration<br>• Elevate local/community level knowledge, reflexively consider HIC knowledge |
| | **Knowledge production** | • Decentralize knowledge platforms<br>• Increase community engagement, prioritize local, lived experience<br>• Centre Indigenous ways of knowing<br>• Engage diverse groups in decision making towards knowledge production<br>• Increase representation in academic journals |
| Shift paradigm in how we **think** about global health | **Knowledge history** | • Identify and acknowledge how structures such as colonialism, racism, etc., pose a threat to health equity<br>• Ground analysis in colonialism and its intersections with other structures (i.e., white supremacy)<br>• Institutional reckoning to mitigate harms perpetuated by structures on which they were built |
| Shift how we **think about and do** global health | **Knowledge justice** | • Diversify what knowledge is considered valuable (ex., outside of metrics)<br>• Shift focus of knowledge and actors in global health to local (disrupt epistemic injustice)<br>• Prioritize reciprocal flows and critique of knowledge<br>• Create new learning platforms and knowledge legitimacy outside of academic English |
| | **Reflexivity** | • Default to local gaze, instead of Western gaze<br>• Critically consider and reflect on individual and institutional positionality, behaviour, unconscious bias<br>• Recognize basis of lens and framing<br>• Listen deeply, listen differently<br>• Dignity based practice |

A significant aspect of the conceptual framework is the idea of "shifting to the local" which is important to briefly interrogate. Contractor & Dasgupta [58] discuss the limitations of simply shifting money and power to low and middle income countries, or "the local", and assuming that contexts are neutral or not fraught with internal power structures (i.e., caste). However, many scholars argue that although it is challenging and not as simple as just "shifting to local", HICs should not remain the centre of power and decision making [46, 59]. As such, this framework suggests a re-centering/redistribution of power where there is greater mutual accountability, collaboration and value prioritized for non-Western ways of knowing, leading and funding. Note, the use of the term "we" is not intended to imply agreement about this potential future for global health and who/what constitutes the "we" will change across different potential futures for the field.

## Implications & benefits

By understanding how the FIAP influences ASRH projects, this paper aims to inform future funding and related practices by the Canadian government to strengthen alignment with the

priorities of the FIAP, NGOs, communities, and individuals involved in ASRH projects. This study offers a potential path of action, from the perspective of the implementing actors, to shift the practice of funding and supporting ASRH projects. As such, this study contributes to the information available to practitioners, researchers, funders, and policy makers to foster more equitable global health practice.

## Methods

Key informant interviews were conducted to investigate stakeholder perspectives about the FIAP and how the policy influences ASRH projects in Eastern sub-Saharan Africa. Both qualitative description (QD) and stakeholder analysis recommend the use of semi-structured interviews for focused data collection [60–65].

Stakeholder analysis is a process of systematically gathering and analyzing information to determine stakeholder interests, interrelations, intentions, and roles in policy change [62–64, 66–68]. What, or who, constitutes a stakeholder has been widely discussed in literature on the approach [62–65, 68–70]. In general, a stakeholder can be defined as individuals, groups or organizations that share common interests and holds interest in the outcomes of certain decisions or objectives [62, 63, 65, 68]. Stakeholder analysis offered an effective structure to engage and recruit key informants from NGOs which are part of a specific stakeholder group described by Schiller et al. [65] as civil society organizations (CSOs). According to Schiller et al. [65], CSOs include NGOs, faith-based organizations, and "Indigenous/ethnic groups" (p. 5). The Canadian government has identified NGOs specifically, as essential actors in implementing the FIAP. However, there is limited information about the perspectives of key informants within NGOs on the FIAP and its influence on ASRH projects.

### Ethics and informed consent

This study was reviewed and approved by the Hamilton Integrated Research Ethics Board (HiREB #13761). A letter of information and consent was shared with participants prior to the interview via email and if interested, participants provided written consent by signing the letter of information and consent. Oral consent was also gathered from participants prior to the start of the interview. All participation was voluntary. Participants were made aware that interviews would be conducted confidentially and information would be immediately de-identified upon completion of the interview to maintain privacy. Participants were also made aware that there were no foreseeable consequences to any future GAC funding.

### Sampling and recruitment

To identify key informants from NGOs working on ASRH projects funded through the FIAP, the lead student investigator (HC) met with a colleague from the Canadian Partnership for Women and Children's Health (CanWaCH). CanWaCH is an organization that aims to collate the knowledge and expertise of Canadian NGOs to advance the health and rights of women, children, and adolescents globally [71]. To this end, CanWaCH has developed a "Project Explorer" database outlining all GAC funded projects [22]. As such, the lead investigator completed an initial search of key informants through the CanWaCH Project Explorer database. Key informants were included in the study if they were individuals in Canada leading ASRH projects implemented in Eastern sub-Saharan Africa, that were at least 90% funded by GAC from 2018 onwards. These individuals were connected to the country-based implementing organizations so could speak to the wide scope of FIAP influence.

After this list of potential key informants was developed, the lead student investigator worked with her colleague at CanWaCH to gather contact information for each individual.

Based on recommended procedures for stakeholder analysis and QD [63, 65, 70, 72], key informants were then purposively recruited via email. It was made clear in the initial email to key informants that CanWaCH would not be involved in the interviews and participating in an interview would in no way impact partnership with CanWaCH or the funding received from GAC.

## Data collection

Of the eleven key informants that fit the inclusion criteria and were purposively recruited between 27-02-2023 and 30-04-2023, eight individuals participated in interviews between 10-03-2023 and 06-05-2023. A framework of questions for the interview guide was informed by stakeholder analysis tools [62, 63, 65] and qualitative interview methodology [73, 74]. Content for the interview questions was developed from the study research question, objectives, and previous analyses of the FIAP and ASRH research by Chidwick et al. [35]. The interview guide consisted of 13 pre-determined open-ended questions and additional probing questions. Questions asked about the key informant's work and role at the organization, details on the ASRH project of focus and how it was developed and implemented, and their perspective on the FIAP and its connection to the ASRH project (S1 Text). This structure intended to encourage a conversational style.

The lead student investigator conducted the interviews in English over the Zoom platform. Interviews took about 50 to 60 minutes, where all participation was voluntary and confidential. Recommended practices for conducting ethical remote interviews, including the use of the Zoom platform specifically, flexibility with possible technical and connectivity challenges, and sharing the privacy policy from Zoom with participants in the informed consent process, were consulted and continually considered throughout the process [75–77]. Interviews were recorded, de-identified and transcribed to ensure confidentiality and effective data analysis. All information from the study was stored in electronic encrypted and password-protected folders.

## Analytic strategy

Key informant interviews were analyzed through a QD approach using content analysis. QD is a research methodology that aims to describe experiences, events, and perspectives in a factual and authentic way, through concurrent data collection and analysis [72, 78–80]. QD studies explore the who, what, and where of events or experiences, based on a constructivist paradigm which notes perspectives are subjective and specific to social, cultural, and historical context while aiming to be inclusive of all perspectives regardless of existing social power hierarchies [72, 78, 79]. Aligned with Sandelowski's view of QD, this study emphasizes the value of participant's perspectives as viable end-products [72, 79]. QD has also been noted as especially effective in health research, providing an opportunity for factual responses from participants and description of those responses to understand the impact of processes and policy [78]. In this study, QD offered an opportunity to describe key informant perspectives of the FIAP which have not been previously captured.

Data analysis was completed concurrently with data collection so, interview transcripts and documents were analysed together to describe the phenomena of interest with limited integration of external theory or interpretation [72, 78–82]. Data was explored inductively to identify and describe recurring themes, concepts, and patterns [72, 78, 79, 83]. Recordings from participant interviews were immediately de-identified using a study key and transcribed by the lead student investigator. Transcripts were then coded through the NVivo 12 software, enabling transparency, flexibility, and trustworthiness of the data [84].

A codebook was developed inductively from interview data. Codes are action-oriented words or labels used to describe parts of interview text, reflecting recurring themes or topics [78, 82]. The study codebook was developed through a process of line-by-line coding and clustering of emerging patterns. Line-by-line coding is the process of identifying words and sentences that appear to capture thoughts on concepts, allowing for open, inductive themes to emerge [78, 81]. One member of the research team (HC) line-by-line coded six interview transcripts and another member of the team (LS) coded two transcripts. After this, the team discussed the line-by-line coding and codes were grouped into categories and clustered into larger sections based on their area of focus. Codes were further synthesized and refined, and emerging themes were noted (S1 Table). These themes included GAC guidelines, consultation with partners, challenges with implementing projects, tensions between the FIAP and project goals, along with opportunities for future funding. Themes from the codebook were reviewed for consistency against a set of government documents such as a Call for Proposals, Results-Based Management Framework and Gender Analysis Framework (S2 Text). This provisional codebook and an overview of emerging themes was also shared, via email, with interview participants who indicated interest in providing feedback on the codebook during the interview. Based on feedback from participants stating they were content with the codebook and continuing discussions with the research team, the codebook was revised as needed. Relevant data was re-coded to ensure consistency and quality of the analysis. After inductive analysis of the data and identification of themes, results were explored in relation to the conceptual framework for an imagined future for global health proposed by Chidwick et al. [35].

Memoing by the lead investigator was completed throughout the process of study development and data collection to ensure researcher reflexivity and capture the lead investigator's stance and process for considering data [78, 85]. Reflective memos also assisted in ensuring the integrity of participant perspectives, as researcher perspectives and biases were identified and the analysis was then able to stay grounded in the data [78, 80, 85].

## Results

The findings presented below are organized into five main sections: an overview of key informants and their ASRH projects; the significance of partnerships in ASRH projects; the influence of the FIAP; tensions between the FIAP, GAC structures and context of implementation; and what participants described as push for change. These findings are subsequently discussed in relation to the conceptual framework for an imagined future for global health in the Discussion section of this paper.

### Key informants & ASRH projects

Eight key informants were involved in the study, five who used she/her pronouns and three who used he/him pronouns. Majority of participants (n = 6) were Program Managers or Program Officers at their organizations, with a specific focus on SRHR. One participant was a Gender Equality Advisor at their organization, and one participant was the SRHR Director. Participants described their role at Canadian organizations and in the SRHR projects as leading on management and coordination of projects from Canada through collaborative, non-hierarchical and accompanying approaches (see Table 2). Participants noted in particular that they lead on communication and compliance with GAC and emphasized their role as the intermediary between GAC and partner organizations, to *"soften the blow from. . . GAC compliance"* (Study Participant). Note, participant quotes are referenced using a generic indicator "Study Participant". Information has not been disaggregated based on participant ID's in order to ensure privacy and confidentiality. Due to the small number of participants in the

**Table 2. Overview of key informants and ASRH projects.**

| Org. ID | Time-frame | Project Goal | GAC Funding |
|---|---|---|---|
| 1 | 2020–25 | Help adolescent girls and young women access better SRH care | < $20mil |
| 2a | 2021–27 | Improve access to high quality, gender sensitive SRH services for adolescent girls and young women | < $20mil |
| 2b | 2021–27 | Improve access to high quality, gender sensitive SRH services for adolescent girls and young women | < $20mil |
| 3 | 2021–26 | Help adolescents access better SRH care | > $20mil |
| 4 | 2022–2027 | Support adolescents and existing NGOs provide services to access SRH care | > $20mil |
| 5 | 2021–28 | Improve SRHR of adolescent girls, particularly girls who have dropped out of school | > $20mil |
| 6 | 2019–2024 | Improve access to SRH services for adolescents | < $20mil |
| 7 | 2021–2028 | Improve access to ASRH care | > $20mil |

study, participant privacy was a significant priority. Disaggregating participants by their ID could lead to identification if patterns of language, through the quotes shared, are familiar or recognizable. As such, a generic indicator has been used throughout. We aimed to include as many participant voices as possible.

Projects were included if they were 90% funded by GAC (or more). To protect participant privacy, we used $20million as the general indicator to demonstrate funding in Table 2, as most projects received funding that was just below or just above $20million. GAC funding was described as distributed mostly to in-country partners that are responsible for the implementation of the ASRH projects. Fig 1 outlines the structure of funding, identifying where individuals from Canadian organizations are involved (also see S1 Fig).

Projects were implemented in seven Eastern sub-Saharan Africa countries including, Malawi, Mozambique, Uganda, Democratic Republic of the Congo, Burundi, Kenya, and Zambia. Many projects were implemented in more than one of these countries. Projects were usually structured through a three to four pillar model that included elements such as health systems strengthening, access to care and SRH information, quality of care, and advocacy. Some organizations had existing approaches to SRHR that were noted as *"very progressive. . . inclusive, [and] intersectional"* (Study Participant), whereas other organizations discussed their SRHR focus as newer (within the last 2–3 years). Participants also discussed prioritizing projects that focused on the four neglected, or "under-funded", areas of SRHR as defined by Canadian government, which include family planning and contraceptives, advocacy, safe abortion and comprehensive sexuality education (CSE) [16].

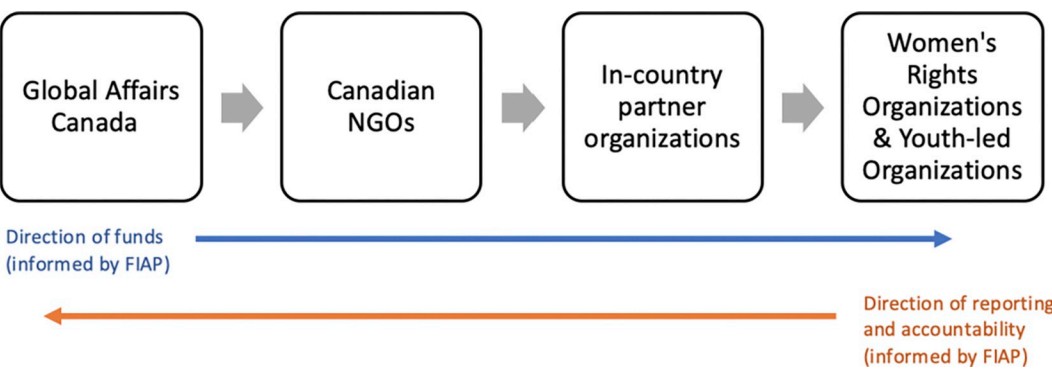

**Fig 1. Structure of funding and reporting.**

## Partnerships

In the development stage of projects, participants described consultation with partners as an important part of the process. Partners included in-country NGOs, international NGOs, women's rights organizations (WROs) and youth led organizations (YLOs). In-country and international NGOs were usually larger organizations whereas WROs and YLOs were smaller organizations. In the implementation stage of projects, participants discussed partner organizations as the primary implementers. One participant noted the importance of prioritizing partner organizations stating, *"we always have to think about our partners, the people that we're working with, and so. . . it's a balance of where we want to go and who we can support"* (Study Participant). Evaluation was largely described as being led by partners, with Canadian organizations as a support actor, but many participants noted GAC compliance requirements were difficult to navigate, especially for YLOs and WROs as many smaller, grassroots organizations do not have the capacity to meet the strict GAC auditing and reporting requirements. Opportunities for more partner led, participatory evaluation approaches, where outcomes are developed in collaboration with partners rather than solely through funding requirements, such as outcome harvesting (a participatory and iterative evaluation process that involves collecting evidence of change and then working backwards with stakeholders and end-users to understand how a project or program contributed to that change) were noted by different participants although limitations were also noted. These limitations which included differing values in the project than those in-country are discussed in greater detail in the section below about tensions.

> *"there is an initiative to, to go beyond the traditional, you know, baseline, midline, end-line, quantitative survey, and sort of qualitative focus group approach to include some outcome mapping, maybe a little bit of outcome harvesting"* (Study Participant)

> *"when we're talking about our feminist monitoring, evaluation, accountability, learning processes, we want our partners to be in the driver's seat, we want them to really be able to, like, say, here are our priorities, here's what we're working on it. Um, but we acknowledge that there's limitations there. You know, some of our partners are really uncomfortable with the idea of abortion. So. . . there is a tight balancing act"* (Study Participant)

In general, a range of participants described project partnerships as both the project's main strength and main challenge or weakness. For example, for one participant project strengths included the *"dynamic consortium [that] everybody contributes to it differently"* (Study Participant) and for another, the *"dedicated and knowledgeable staff who are really trained and are committed to providing care at whatever time"* (Study Participant). Challenges with partnerships included the quality of consultations in proposal development due to short submission timelines (6–8 weeks) along with difficulty coordinating and communicating with many partner organizations. For example, a few participants described extensive requirements from GAC to track spending and budgets across the project along with detailed reports which were time consuming.

Different participants also described operational challenges with the process of engaging adolescents as collaborators and partners. These challenges included navigating parental consent while respecting adolescent privacy, the ethics of safely engaging adolescents, and mitigating how much safeguards become barrier to engaging adolescents.

> *"every time you talk about participation of children and youth, you all of a sudden become so sensitive, so careful, that nothing happens. . .So you're too cautious to engage the community,*

*and at the end of the day to do it in a perfect way you are basically missing the point of doing it at all"* (Study Participant)

*"it's a bit tricky working with adolescents because of parental consent, because of ethics. . .the constraints of those Call for Proposals and the timelines makes it really difficult to engage meaningfully with young people at the, at the proposal development stage. Once we get to, once a project has been accepted, there's a bit more time and space to engage young people."* (Study Participant)

## Influence of the FIAP

Participants described that the FIAP significantly influences ASRH projects through funding and determining project priorities. For example, one participant stated that their organization has to *"go where the money is"* (Study Participant) and develop projects based on the current government priorities (i.e., SRHR). Other participants noted that the FIAP *"certainly shapes [the] trajectory or it creates the opportunity to take on topics that [are] considered neglected"* (Study Participant) and, *". . .why [the project] was funded or approved is because, you know, whatever we put forward was bound to be in alignment with FIAP and other GAC priorities"* (Study Participant). Further a participant noted that *". . .the Call for Proposals outlined what the intermediate outcomes had to be. So it was like, certainly there is a major focus from the donor around what the project was supposed to look like. Obviously, there's room to adapt within that. . ."* (Study Participant). The FIAP was also described as informing the Call for Proposals, which in turn shapes how the project is designed, what the priorities of the projects are, and its outcome measures.

Similarly, different participants noted that aspects of the FIAP such as gender transformation and a focus on the four neglected areas of SRHR shaped ASRH projects. For example,

*"the whole project approach is very much geared towards being gender transformative. . .. what are the types of interventions? How is it being evaluated? It's very much like cross, cross-cutting and throughout everything"* (Study Participant).

*"we answer to a call to applications. . . For the past, maybe two, three years, it's a lot related to the neglected field. . . so there's safe abortion and post abortion service, complete education, a complete sexual education. . ."* (Study Participant).

In addition to the FIAP, participants noted that existing organizational values, experiences and approaches to SRHR that prioritized feminist approaches, and comprehensive SRHR were important in shaping projects.

## Tensions

When discussing the FIAP, majority of key informants shared the positive impacts of the policy conjunctionally with its limitations. Thus, two significant tensions of the FIAP and its implementation through ASRH projects emerged from key informant perspectives. First, the tension of the FIAP as an important policy although difficult to implement within GAC processes and structure–the "operational" tension. Second, the tension of the FIAP as a progressive and gender transformative policy although not aligned with context-specific ideas of ASRH–the "contextual" tension.

**Operational.**    The first tension draws on the incongruity between the policy goals of the FIAP and the funding requirements from GAC. The FIAP is described as *"a huge step forward*

*in an approach to international cooperation by the Canadian government. . . relatively speaking, very, very progressive"* (Study Participant). On the other hand, GAC processes are described as *"strict"* and prohibitive, limiting the potential to operationalize the FIAP. For example, participants said,

> *". . .to a certain extent, [the FIAP] was sort of, in some ways, putting the cart before the horse, because. . .GAC was not ready to action a lot of what this policy demands, right, it wasn't prepared. It's great, though. . .the impetus. . . So even as the operationalizing of the policy needed to catch up and has been catching up, it's still not there, but has been catching up."* (Study Participant)

> *"I think for us the problem is not the FIAP. The problem is like the machine–the government, in terms of their structure, their ways of reporting, their ways of, you know, functioning and working and their requests and how they request it, and just the way they disperse the money or not. . ."* (Study Participant)

> *". . .it's great that the FIAP exists, it's great that we have a focus on SRHR, that we have a focus on contested areas and neglected areas of SRHR. But the ways of working at GAC are prohibitive, and like extremely limiting in terms of what we can do, right, like, ideally, we would have really participatory processes that put youth in the driver's seat that, you know, recognized youth agency and their, you know, evolving capacities as young people to make important decisions about their sexual health. . .But in terms of like engaging [youth] on the kind of like operational side. . .if you want youth led organizations to meaningfully participate in these types of projects, you need to understand that grassroots women's rights organizations, youth led organizations do not have the structures typically, to really be able to produce solid financial reporting, be ready for an audit, provide all the reporting that you require."* (Study Participant)

This operational tension is also related to challenges that participants describe, namely, with timelines. For example, one participant described a timeline of 5 years being too little to implement a project in a respectful, context-specific way as the opportunity to build partnerships and trust is not included in the timeframe.

**Contextual.** The second tension draws on the misalignment between the content of the FIAP and the context where ASRH projects are being implemented. Participants described contradictions of the FIAP in promoting empowerment and capacity building but then also prescribing a set of feminist values or certain idea of what SRHR is, leading to questions about the role of Canadian organizations in ASRH projects. For example, participants said,

> *"So when you bring it down to like, what you need to do as a country, that contradiction, that, in our logic, in our template of development, it makes sense to tell them this is the best thing to do. But at the same time we are talking about their empowerment. Like, if I am talking about your empowerment and saying to you this is the right way to do, how you are involved?. . .Is that empowerment or a temporary kind of supply of information and services? So when we work in the sector, it's, it's even specific to this project, it's very difficult to accept both scenario. Neither we can accept that women are humiliated and harassed, and whatnot in a country context, whereas, taking a feminist international assistance policy and pushing them to do it. Both are wrong"* (Study Participant)

> *". . .community priorities often conflict with FIAP. It is not a straight solution like oh, so let's only do the community things. We even don't know what the community wants, engaging them earlier, is easier said than done"* (Study Participant)

"*Like when we talk about abortion and LGBTQ plus rights. That's, that's a whole other story. And then again, we don't want to impose our views.*" (Study Participant)

Projects implemented in some East African countries were noted to have legal frameworks that somewhat aligned with the FIAP, making it easier or less risky to implement project aspects (i.e., inclusion of LBGTQ communities, CSE). Contrarily, in other East African countries, FIAP values were described as conflicting with national legal frameworks and/or cultural values, making it more prescriptive and sometimes dangerous to implement.

To address this contextual tension, participants noted the importance of moving forward with awareness and with humility in building trusting and respectful partnerships with organizations and communities. Different participants said,

"*. . .building it in and being cognizant of local realities and perspectives of like the need to build that trust. And kind of the generational time frames of, of the change. So it's like it's important that things don't feel like they're being, being imposed in any way, and so, be designed with that in mind where it's, it's not, it's not prescriptive of what needs to be implemented, but rather it's a two way conversation*" (Study Participant)

"*I think where we land is that we take a rights based approach, and that people have rights, over their bodies, over their lives. And that's where we come back to so we can accept that, you know, there's different values that exist within different people and organizations, but at the end of the day, if people want to work on an SRHR project funded by GAC, with Org X, we want everyone to, to understand that we take a rights based approach to it. So they can have their personal views. And also respect that other people have human rights that are worthy of being upheld, and promoted and supported.*" (Study Participant)

## Push for change

When participants were asked about what policy makers and funding organizations should prioritize in supporting ASRH projects, participants did not mention changes to the FIAP. Rather, most participants named opportunities for an operational push for change amongst government processes for funding. First, participants emphasized the importance of flexible timelines and funding throughout projects by stating,

"*So my specific recommendation, if I made it that to the Government of Canada is that you have to allow flexible funding. . . give some flexibility of the funding*" (Study Participant)

"*. . . longer project cycles or opportunities for longer project cycles. . . and, yeah having those flexible, flexible pools of funding*" (Study Participant)

Similarly participants discussed the importance of longer timeframes to submit Calls for Proposals.

"*What can we do in three weeks, really, and then we have to go there and then do consultations. And there's a limit to what we can do. But. . .if we had like something like six months, or like a year, and then it would, the projects themselves would be so much better, will be more detail would be more innovative*" (Study Participant)

Second, participants called for GAC to be bold and take more risks in supporting innovation in SRHR.

"...*invest in progressive SRHR. It's worth it. It's the way forward that empowers diverse groups of adolescent girls and young women the most. It really speaks to their needs, in their context*" (Study Participant)

"...*be bold, they're always like, we want to innovate... well, innovation comes with risk. So [GAC has] to be willing that in our reporting, the project might have failed*" (Study Participant)

Third, participants described accountability and security as important for policy makers and government actors to prioritize when funding progressive SRHR. This was especially noted in contexts where FIAP values such as supporting diverse sexual orientations and gender identities do not align with the in-country values which puts individuals who work on FIAP funded projects at risk.

"...*security, is very, very, very important... So I'm talking about youth here, but it could be also activists, if you're being active, like activist on abortion, for example, it might be at some point you need to be extracted from the country, you know?*" (Study Participant)

"*It's like SOGI [sexual orientation and gender identity] issues in Uganda... that's a real head scratcher for me at the moment... we really want to support Ugandan civil society, who are in a really difficult place right now. And we want to, like amplify their messages, in terms of saying that the proposed legislation in Uganda is very problematic...the challenges, though, for Org X in Uganda is that if they go around saying this stuff that they put their staff and their partners at risk, and so there's significant security restraints and considerations to take.*" (Study Participant)

Finally, participants noted the importance of GAC ceding power and control while emphasizing reflection and listening in how GAC funds ASRH projects and what this funding means.

"*I think also, to not be afraid to cede a little bit of control and power. Right. That's, I think, it's sort of, that's even taking a step further back. Right. Don't be afraid, don't be afraid. Friends in compliance at GAC, and I know they are undergoing this major sort of overhaul, whatever, we'll see how major it is, but don't be afraid... So I think that would be an important thing for policymakers and decision makers to at least think about for five seconds. What does it mean? What reparations mean? What reparations mean vis a vie aid?*" (Study Participant)

The Women's Voice and Leadership (WVL) funding call was also discussed by a few participants as an effective approach to funding going forward. Participants discussed WVL funded projects as having more flexibility in funding and reporting so that partnerships with grassroots organizations, WROs, and YLOs are more feasible and effective.

## Discussion

Findings suggest that the FIAP influences ASRH projects in Eastern sub-Saharan Africa in impactful and positive ways although there are contradictions and tensions that arise. These tensions indicate opportunity for changes to funding and related processes by GAC to more effectively support ASRH practice that upholds feminist and multi-sectoral approaches promoted in the FIAP and that meaningfully aligns with the conceptual framework for an imagined future for global health as proposed by Chidwick et al. [35]. Findings are discussed in relation to the key areas of the conceptual framework for an imagined future–funding, leadership, knowledge (production, history, justice), and reflexivity.

## Funding

Although informed by the FIAP and its progressive focus on SRHR, GAC funding was described as strict and prohibitive. Specifically, despite its intended framing, the funding structure of the FIAP through GAC was described as a barrier to operationalizing feminist, sustainable approaches to ASRH. The conceptual framework for an imagined future calls for a redistribution and decentralization of resources and funding to the "local", which refers to the people closest to the work [35]. In addition, the conceptual framework outlines funder accountability to the local grantees along with greater alignment between funding and local priorities [35]. Findings suggest that these calls to decentralize and redistribute funding to the local are not currently being met in ASRH projects given the restrictive nature of GAC funding and inaccessibility of GAC financial reporting and auditing for grassroots organizations. This includes the extensive reporting requirements based on pre-determined key performance indicators and in strict alignment with projected budgets, along with short timelines to implement project activities. Current Canadian funding for organizations is led by "direction and control" provisions from Canada's Income Tax Act which requires Canadian organizations to provide oversight and operational control of funds [86]. The Canadian Council for International Cooperation (CCIC) [86] notes that this structure limits the ability of partner organizations to adapt to their context due to unnecessary bureaucratic burdens and constrained decision making for partners due to power dynamics. As such, alternative funding structures and oversight processes have been suggested to emphasize flexible funding and reporting to allow for adaptation of approaches to local contexts and needs [86, 87]. As noted by participants, the WVL Program is a step forward in flexible funding, despite its challenges [87]. In particular, the WVL Program funds WROs directly, working to mitigate some of the challenges of direction and control from Canadian organizations as financial and technical intermediaries. In order to meaningfully operationalize feminist, sustainable approaches to ASRH, as called for in the FIAP and which align with the conceptual framework for an imagined future, GAC is encouraged to shift funding structures to foster flexibility, adaptation, and context-specificity through local leadership.

## Leadership

Results indicate that collaborative, non-hierarchal partnerships along with cross-context learning were priorities and strengths of ASRH projects that align with feminist, multi-sectoral approaches, encouraged in the FIAP. However, the operational tension of the GAC "machine" (i.e., reporting requirements, timelines etc.) limited how meaningful and collaborative these partnerships were. Although collaboration is promoted in the FIAP, GAC processes lack support for alternative approaches to consultation and partnership. The 2023 Auditor General's Report also presented potential constraints to partnerships by further encouraging a tightening of reporting and regulatory processes to better show the impact of Canadian funding [88]. The conceptual framework for an imagined future outlines shifts in leadership such as reorienting decision making to the local and decentralizing power while also encouraging South-South collaboration [35]. Findings suggest that there is opportunity to shift structures of leadership, and thus reporting, to be more feminist and flexible, and extend project timelines to strengthen the ability of Canadian organizations to build sustainable and impactful relationships with partner organizations. This aligns with growing initiatives amongst NGOs to redefine measures of success and reorient who has agency to tell the story of impact and how this story is told in development and global health projects [89, 90]. For example, initiatives such as the Pledge for Change 2030 and CSO Partnership for Development Effectiveness advocate for equitable partnerships, accountability and authentic storytelling in development and global

health which supports opportunities to decentralize leadership and ultimately, more effectively align with the FIAP [90, 91]. In addition, Global Health 50/50 Reports in 2022 and 2023 identified continued underrepresentation of women in global health leadership roles, especially from low income countries, calling for board and organizational leadership to adopt "Nothing About Us Without Us" approaches to inclusion and engagement [92, 93].

Further analysis of the results in terms of partnerships, prompts additional consideration about the role of power in leadership and project structures. Although many participants described their relationships with partner organizations as non-hierarchical, the operational tensions that arise suggest that GAC holds structural power, limiting the opportunity to meaningfully operationalize feminist approaches, as encouraged in the FIAP. Structural power is power to shape and determine how things are done–power that is less visible [94, 95]. For example, GAC holds structural power to shape the behaviours and processes of NGOs through aspects such as a Call for Proposals and reporting compliance guidelines. This power is also directional, where accountability is from the partner/Canadian organizations to the Canadian government. The conceptual framework for an imagined future calls for "funder accountability to the local" [35]. This idea has been supported in other disciplines such as Global Development studies where scholar Shama Dossa describes directional accountability through processes such as reporting as a mode of surveillance [96]. Dossa [96] further asks, "What if we trust instead of surveil?" indicating a potential way forward in shifting power from GAC to partner organizations and creating opportunities for mutual accountability which more strongly aligns with the FIAP.

## Knowledge

The contextual tension that arose from results indicates misalignment of implementing the FIAP through ASRH projects with aspects of an imagined future, namely, knowledge production, knowledge history and knowledge justice. Results indicate that community priorities sometimes conflict with the FIAP. For example, participants described conflicting views with partner organizations for some SRHR concepts such as abortion, while also acknowledging that it was wrong to impose certain (Western) views. The knowledge production, history and justice aspects of an imagined future call for decentralizing knowledge platforms, prioritizing local and lived experience, and identifying how structures such as colonialism impact what knowledge is valued towards diversifying these values [35]. Recently, a few scholars have begun to advocate for dignity-based practice [97, 98], providing a potential path forward to work within these contextual tensions. Dignity-based practice recognizes that communities at the centre of the challenge are best positioned to define their own needs [98]. Abimbola [97] further notes that "We [in global health] mustn't give in to the notion that the global South must come to the global North's table to be seen as knowers" (p. 2). Similarly, and as noted by some participants, rights-based approaches that enable individuals and particularly women and girls to actively take part in SRH decisions [99], also provide a path forward for how to balance differing contextual values.

Findings about the contextual tension between the knowledge promoted in the FIAP and in-country knowledge and values, also prompts a discussion about a duty of care. Health care providers in Canada hold an ethical and legal duty to provide care [100, 101]. This includes providing both information and access to safe, compassionate, competent and ethical healthcare [100, 101]. Considering a duty of care when implementing the FIAP, leads to the question of, if funding from GAC is received, is there a duty of care to employ human-rights based approaches to SRHR regardless of contextual views? And further, if there is a duty of care, what responsibility does GAC have to protect the security of individuals engaging in SRHR services such as abortion that could put the service providers and patients at risk? Currently, the Canadian government follows

the Voices at Risk guidelines to protect human rights defenders abroad [102]. The guidelines establish that the Canadian government supports individuals who defend human rights and aim to ensure they are able to do their work in a safe environment [102]. As such, GAC evidently has a responsibility to protect individuals who advocate for human rights in contexts where FIAP values such as feminist approaches and diverse sexual and gender identities, are not supported. Going forward, given the likelihood of conflicting views, flexible processes for consultation, a commitment to rights-based, dignity-based care, and a commitment to support the security of individuals working in this area, through existing Voices at Risk guidelines, should be prioritized. By prioritizing such processes, greater alignment with the conceptual framework for an imagined future, particularly in terms of knowledge, is possible.

## Reflexivity

Humility and processes to build trust with partner organizations and communities were also highlighted as a way forward through operational and contextual tensions that arise when implementing ASRH projects through the FIAP. Humility and building trust are values that align with calls for reflexivity in the conceptual framework for an imagined future [35]. For example, participants called GAC to cede power and control while supporting organizations in building trust and respectful partnership, and practice reflection on what aid means generally and in terms of reparations for potential historical power dynamics and colonialism. Reflexivity in the conceptual framework for an imagined future advocates for critically considering and reflecting on both individual and institutional positionality, recognizing lens and framing, and listening deeply and differently [35]. Feminist and dignity-based approaches similarly advocate for reflexive practice through participatory methods such as photovoice and body-mapping that integrate intersectionality and reconceptualize what knowledge is seen as legitimate (i.e., epistemic justice) [98, 103]. Photovoice is a participatory, capacity-focused method that uses images to capture experiences and facilitate critical discussion and learning [104–106], whereas body-mapping engages participants to draw an outline of their body and through guided reflection visually write or draw their experiences of certain topics [107, 108]. Both are rooted in justice and empowerment, and create opportunities for critical discussion of power and history. Evidently, there is continued opportunity to re-imagine GAC processes to foster a flexible structure of consultation that supports humility, building trust, and reflexive practice and thus greater alignment with the FIAP.

## Considering the push for change and an imagined future

Findings indicate that there is discussion within NGOs to shift processes towards thinking and doing global health differently. Key informants note opportunities for operational changes from the Canadian government to more effectively support ASRH projects. Fig 2 explores these opportunities for shifts in relation to the conceptual framework for an imagined future [35]. Opportunities suggested by participants significantly align with calls for shifts in leadership from HIC institutions to local, along with increased reflexivity amongst institutions and HIC individuals to mitigate harmful power dynamics. The call for GAC to cede power and control and integrate greater reflection aligns with most aspects of an imagined future. Based on the alignment of the FIAP with the conceptual framework for an imagine future, these shifts from GAC would strengthen the potential for feminist, sustainable approaches to ASRH.

## Limitations

Important limitations to note include, first, challenges with completing this study, which was part of a doctoral dissertation project, during the COVID-19 pandemic. The pandemic

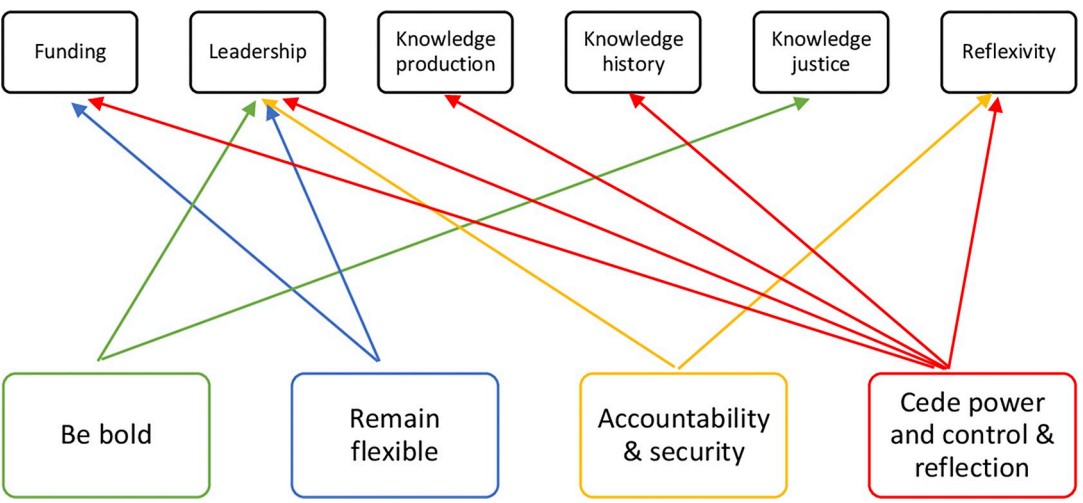

**Fig 2. Alignment of an imagined future with participant future directions.**

affected the capacity of organizations to engage, and ability of the lead investigator to build trust with organizations before interviews began remotely. Building trust and relationships was particularly important in this research given the complexity of the topic–SRH. Relationships and rapport with organizations involved were developed gradually and through existing connections with CanWaCH, to maximize the opportunity for meaningful interviews. Flexibility in the timeline, recruitment, and remote interview approach was also prioritized to accommodate for the changing capacity of organizations. This resulted in most interviews being rescheduled due to participant capacity. Second, the scope of NGOs involved in this study is limited based on the inclusion criteria and the limitations COVID imposed on the lead researcher's ability to engage organizations in multiple foreign countries (i.e., processes of ethics approval, relationship building abroad etc., was limited throughout the dissertation and related studies). To ensure feasibility of the study, key informants from NGOs in Canada were involved in interviews resulting in a small sample size. This sample size somewhat limited the diversity and breadth of perspectives involved however included valuable perspectives from Canadian NGO key informants who work most directly with the Canadian government through SRH projects and whose perspectives on the FIAP had yet to be captured. Although the perspectives from country-based NGOs were minimal, Canadian organizations work directly with country-based organizations and were able to engage meaningfully with the interview questions while reflecting on how the FIAP shapes ASRH projects in general. In addition, data from interviews was triangulated with data from the relevant government documents (S1 Text), which enhanced the rigour and quality of the discussion from this study, ultimately aiming to influence the Canadian development sector's approach to ASRH projects. Further research on the topic with country-based NGOs is encouraged and important in future study. Research is also encouraged to prioritize intentional and transparent consideration with partners about how to navigate the political, social, and cultural complexities of working on SRH projects in different contexts.

## Conclusion

This study has explored key informant perspectives on how the FIAP influences ASRH projects in Eastern sub-Saharan Africa and their subsequent alignment with the conceptual framework for an imagined future for global health as proposed by Chidwick et al. [35]. Findings

suggest that the language of the FIAP is aligned with an imagined future for global health although there is opportunity for further action towards operationalizing shifts in GAC funding structures and policy. Specifically, findings highlight opportunity to create more flexible funding processes, expand monitoring and evaluation approaches to include qualitative and feminist ways of measuring impact, promote rights-based and dignity-based approaches to ASRH, and increase timelines for projects in order to facilitate greater consultation and partnership building with communities involved. Future research should explore alternative, flexible processes for funding that foster feminist, participatory and locally-led approaches to ASRH in Eastern sub-Saharan Africa.

## Supporting information

**S1 Text. Interview guide.**
(DOCX)

**S2 Text. Relevant government documents.**
(DOCX)

**S1 Table. Thematic analysis codebook.**
(DOCX)

**S1 Fig. Overview of the funding process.**
(DOCX)

## Acknowledgments

The authors share their gratitude and thanks to the key informants who generously gave their time and insights. Thank you to Imaeyen Okon who supported in participant recruitment and conceptualization. And finally, many thanks to Dr. Erica Nelson for her generous, detailed, and impactful feedback and continued insightful support.

## Author Contributions

**Conceptualization:** Hanna Chidwick, Germaine Tuyisenge, Deborah D. DiLiberto, Lisa Schwartz.

**Data curation:** Hanna Chidwick, Lisa Schwartz.

**Formal analysis:** Hanna Chidwick, Lisa Schwartz.

**Investigation:** Hanna Chidwick.

**Methodology:** Hanna Chidwick, Germaine Tuyisenge, Deborah D. DiLiberto, Lisa Schwartz.

**Project administration:** Hanna Chidwick.

**Supervision:** Germaine Tuyisenge, Deborah D. DiLiberto, Lisa Schwartz.

**Validation:** Hanna Chidwick, Germaine Tuyisenge, Deborah D. DiLiberto, Lisa Schwartz.

**Writing – original draft:** Hanna Chidwick.

**Writing – review & editing:** Hanna Chidwick, Germaine Tuyisenge, Deborah D. DiLiberto, Lisa Schwartz.

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
