## [Decision Letter · Decision Letter 0]

22 Aug 2024

PGPH-D-24-01371

Contradictions and possibilities for change: Exploring stakeholder perspectives of Canada’s Feminist International Assistance Policy (FIAP) and their connection to a future for global health

Dear Hanna Chidwick,

Thank you for submitting your manuscript to PLOS Global Public Health. After careful consideration, we feel that it has merit but does not fully meet PLOS Global Public Health’s publication criteria as it currently stands. Therefore, we invite you to submit a revised version of the manuscript that addresses the points raised during the review process.

Dear Author Team,

Please find Reviewer 2 comments at the bottom of this email. Reviewer 1 comments are attached separately. Regarding Reviewer 2's overarching point, I believe the two components housed within the paper are a strength rather than a weakness. The length is acceptable. But given the concern described by Reviewer 2 about the 'two areas' of the paper, I would recommend seeking out ways to further integrate these areas as able throughout the manuscript. 

Reviewer 2 comments:

To the Authors

In essence I found the paper way too long and really is directed at two areas---how FIAP is operationalized and influences ASRH projects within NGOs in eastern Africa and a precis of Ms Chadwicks dissertation on reimaging global health in a more equitable fashion. My suggestion is to consider splitting this paper into two or concentrate on describing FIAP and some of the difficulties in operationalizing it. I would also expand the number of interviews done. As I read it there were only 8 interviews completed due to COVID issues and these were often done not in the field because of COVID limitations. I found the concept of FIAP Canadian approach fascinating and was unaware of its existence as I am sure many readers outside of Canada would be. For readers not that familiar with this mandate and its impact on NGOs,that in itself makes for an interesting paper—

Specific questions to the line items—

Line 22—What percentage of key informants refused to participate?. Was there possible selection bias that occurred ?What percentage of key informants were queried thru on the ground participants or thru the Canadian NGO office.

Lines 320-326 Paragraph designates 20 million dollars as the cutoff for NGO participation yet in your table there are 4 projects over 20 million.

Lines 353 -356 Important to emphasize the need for outcome parameters to be developed together and not only mandated by Canada funding requirements Define “outcome harvesting” better

Line 480-482 grammar of quote confusing---“Neither we can accept……

Iine 597 What is the restrictive nature of GAC funding. (I know in supplement ) but you might highlight to the reader some of the worst requirements and auditing which the grassroot organizations are having difficulties with.

Line 701-704. What are photovoice and body mapping—define

Line 736-Limitations of COVID and using Canadian NGO partners may have limited the ability of investigators to build trust and get honest answers. Would be helpful to end with looming political issues such as addressing homosexuality, gender violence and other charged subjects

We look forward to receiving your revised manuscript.

Kind regards,

Bram Wispelwey, MD, MS, MPH

Academic Editor

Journal Requirements:

Reviewers' comments:

Reviewer's Responses to Questions

**Comments to the Author**

1. Does this manuscript meet PLOS Global Public Health’s publication criteria? Is the manuscript technically sound, and do the data support the conclusions? The manuscript must describe methodologically and ethically rigorous research with conclusions that are appropriately drawn based on the data presented.

Reviewer #1: Yes

2. Has the statistical analysis been performed appropriately and rigorously?

Reviewer #1: N/A

3. Have the authors made all data underlying the findings in their manuscript fully available (please refer to the Data Availability Statement at the start of the manuscript PDF file)?

Reviewer #1: Yes

4. Is the manuscript presented in an intelligible fashion and written in standard English?

Reviewer #1: Yes

5. Review Comments to the Author

Reviewer #1: Peer Review for PLOS Global Public Health Manuscript: Contradictions and possibilities for change: Exploring stakeholder perspectives of Canada’s Feminist International Assistance Policy (FIAP) and their connection to a future for global health

Title and Abstract: The title accurately reflects the content and focus of the study. The abstract is well-written, providing a concise summary of the study’s objectives, methodology, key findings, and implications. It effectively captures the reader's interest and gives a clear overview of what to expect in the main text.

Introduction: The introduction provides a comprehensive background on the importance of adolescent sexual and reproductive health (ASRH) and the role of Canada’s FIAP. The rationale for the study is clearly articulated, highlighting the gaps in existing research and the need for exploring stakeholder perspectives on FIAP’s influence on ASRH projects. The research questions and objectives are clearly stated and relevant to the topic.

Methodology: The study employs a qualitative approach, using stakeholder analysis and qualitative description through eight interviews with key informants. This methodology is appropriate given the exploratory nature of the research questions. The selection of participants, data collection methods, and analysis procedures are adequately described. However, more details on the interview guide and how thematic analysis was conducted could enhance the reproducibility of the study. Also, the fact that they had to stay within Canada residents makes sense due to the COVID-19 pandemic, but it’s a gap to miss the voices of the implementation partners.

Results: The results section is well-organized and presents the findings clearly and logically. The themes identified from the interviews are relevant and provide valuable insights into the influence of FIAP on ASRH projects. Using direct quotes from stakeholders enriches the narrative and supports the identified themes. The findings highlight the positive aspects of FIAP and the challenges faced in its implementation due to GAC regulations, providing a balanced view.

Discussion: The discussion section effectively interprets the findings concerning the study’s objectives and existing literature. The authors provide a thoughtful analysis of the implications of their findings for policy and practice, particularly the need for more flexible funding processes, expanded monitoring and evaluation approaches, and increased timelines for consultation and partnership building. The connection to the conceptual framework for an imagined future for global health is well-articulated, adding depth to the discussion.

Conclusion: The conclusion briefly summarizes the key findings and their implications. The authors reiterate the potential of FIAP to promote equity and justice in ASRH projects, while also calling for ongoing re-imagination of funding structures and processes. The conclusion aligns with the study’s objectives and provides a clear take-home message.

Strengths:

1. The study addresses a significant gap in the literature by exploring the practical impacts of FIAP on ASRH projects from the perspective of stakeholders.

2. The qualitative approach and use of direct quotes provide rich, contextualized insights.

3. The discussion connects the findings to broader theoretical frameworks, enhancing the study’s contribution to the field.

Weaknesses:

1. The methodology section could benefit from more detailed information on the interview guide and thematic analysis process.

2. The study is based on a relatively small sample size (eight interviews), which may limit the generalizability of the findings.

3. On page 24 there’s a mention of reparations vis a vis aid and there’s not much analysis around that, which is a relevant example of the power dynamic between FIAP and the implementing partners, and how it’s not addressing those, further than the GAC regulations.

Recommendations for Improvement:

1. Include more detailed descriptions of the interview guide and thematic analysis process in the methodology section or as an annex.

2. Discuss the limitations of the small sample size in more detail and consider ways to address this in future research.

3. Would love it if authors could dig deeper on the reparations vis a vis aid comment.

Overall Assessment: This study provides valuable insights into the influence of Canada’s FIAP on ASRH projects and contributes to the broader discourse on global health policy and practice. The findings are relevant and timely, and the manuscript is well-written and structured. With minor revisions to enhance methodological transparency, this article can make a significant contribution to the field, hopefully influencing improvements on how GAC operates and how FIAP becomes more accountable to local stakeholders.

Oriana López Uribe

6. PLOS authors have the option to publish the peer review history of their article (what does this mean?). If published, this will include your full peer review and any attached files.

**Do you want your identity to be public for this peer review?** For information about this choice, including consent withdrawal, please see our Privacy Policy.

Reviewer #1: **Yes: **Oriana López Uribe

---

## [Editor Report · Decision Letter 1]

21 Oct 2024

Contradictions and possibilities for change: Exploring stakeholder perspectives of Canada’s Feminist International Assistance Policy (FIAP) and their connection to a future for global health

PGPH-D-24-01371R1

Dear Dr. Chidwick,

We are pleased to inform you that your manuscript 'Contradictions and possibilities for change: Exploring stakeholder perspectives of Canada’s Feminist International Assistance Policy (FIAP) and their connection to a future for global health' has been provisionally accepted for publication in PLOS Global Public Health.

Best regards,

Bram Wispelwey, MD, MS, MPH

Academic Editor
